# Mixed tin-lead perovskites with balanced crystallization and oxidation barrier for all-perovskite tandem solar cells

Jin Zhou[1], Shiqiang Fu[1], Shun Zhou[1], Lishuai Huang[1], Cheng Wang[1], Hongling Guan[1], Dexin Pu[1], Hongsen Cui[1], Chen Wang[1], Ti Wang[1], Weiwei Meng [2] ✉, Guojia Fang [1] ✉ & Weijun Ke [1] ✉

Mixed tin-lead perovskite solar cells have driven a lot of passion for research because of their vital role in all-perovskite tandem solar cells, which hold the potential for achieving higher efficiencies compared to single-junction counterparts. However, the pronounced disparity in crystallization processes between tin-based perovskites and lead-based perovskites, coupled with the easy $Sn^{2+}$ oxidation, has long been a dominant factor contributing to high defect densities. In this study, we propose a multidimensional strategy to achieve efficient tin-lead perovskite solar cells by employing a functional N-(carboxypheny)guanidine hydrochloride molecule. The tailored N-(carboxypheny)guanidine hydrochloride molecule plays a pivotal role in manipulating the crystallization and grain growth of tin-lead perovskites, while also serving as a preservative to effectively inhibit $Sn^{2+}$ oxidation, owing to the strong binding between N-(carboxypheny)guanidine hydrochloride and tin (II) iodide and the elevated energy barriers for oxidation. Consequently, single-junction tin-lead cells exhibit a stabilized power conversion efficiency of 23.11% and can maintain 97.45% of their initial value even after 3500 h of shelf storage in an inert atmosphere without encapsulation. We further integrate tin-lead perovskites into two-terminal monolithic all-perovskite tandem cells, delivering a certified efficiency of 27.35%.

Metal halide perovskite solar cells (PSCs) possess superb optoelectronic properties compatible with small weight and thin structure, hence promising prospects in the solar energy industry[1–7]. Pure lead (Pb)-based PSCs have demonstrated a record-breaking power conversion efficiency (PCE) of ~26.1%, which is comparable to that of silicon solar cells[8]. However, Pb-based perovskites typically present bandgaps over 1.4 eV, resulting in limited photocurrents. In contrast, mixed tin-lead (Sn-Pb) perovskites with narrow bandgaps (NBGs) of ~1.26 eV play a crucial role in developing efficient all-perovskite tandem solar cells[9–15]. These all-perovskite tandem cells with efficient light harvesting and photon-to-exciton conversion have the potential to surpass the Shockley-Queisser limit of single-junction solar cells and are therefore considered one of the most important pathways to the commercialization of perovskite photovoltaic devices[16–20].

High-quality NBG mixed Sn-Pb bottom subcells are essential for all-perovskite tandem cells, particularly for efficient light harvesting and current-matching with wide bandgap (WBG) subcells in tandem configurations[21,22]. Therefore, it is crucial to utilize thick and high-quality Sn-Pb perovskite films capable of efficiently harvesting photons and generating high current densities. However, NBG Sn-Pb PSCs show

[1]Key Laboratory of Artificial Micro- and Nano-structures of Ministry of Education of China, School of Physics and Technology, Wuhan University, Wuhan, China. [2]South China Academy of Advanced Optoelectronics, South China Normal University, Guangzhou, China. ✉e-mail: wwmeng@m.scnu.edu.cn; gjfang@whu.edu.cn; weijun.ke@whu.edu.cn

inferior efficiency and stability in comparison with their Pb-based counterparts because of the fast crystallization, which would lead to nonuniform nucleation[16,23,24]. The rapid crystallization of Sn per-ovskites causes poor quality of Sn-Pb perovskite films, hindering the fabrication of efficient all-perovskite tandem cells and impeding their commercialization. Although there are reports on customizing the crystallization of pure Pb or pure Sn perovskites, there is a scarcity of studies on mixed Sn-Pb perovskites, presenting a considerable challenge due to the substantial disparity in solvation between Sn and Pb metal ions. Additionally, easy oxidation of $Sn^{2+}$ in Sn-Pb perovskites results in high trap densities and serious nonradiative recombination[25–29]. $Sn^{2+}$ oxidation can occur at various stages, includ-ing before, during, and even after the formation of perovskite crystals, so long as $Sn^{2+}$ is present[30]. To mitigate this problem, the conventional approaches involve creating an Sn-rich environment to counteract Sn vacancies originating from $Sn^{2+}$ oxidation. Specifically, tin fluoride ($SnF_2$) has commonly been employed to introduce an excess of $Sn^{2+}$ ions and prevent the formation of undesired phases[25,31,32]. Alongside $SnF_2$, Sn metal powders could also be introduced to create an Sn-rich environment and facilitate the conversion of detrimental $Sn^{4+}$ back to $Sn^{2+}$ in the perovskite precursors[30]. However, these routes have limited effectiveness as Sn-Pb perovskite films are still prone to oxidation both during and particularly after the preparation process. The inherent oxidation can engender a proliferation of defects and inefficient photon-to-exciton conversion within the perovskite films. Given that unbalanced crystallization and spontaneous $Sn^{2+}$ oxidation are the primary challenges in mixed Sn-Pb PSCs and their tandems, it is imperative to develop effective strategies to overcome these issues and drive their further advancement.

Here, we propose a systematic and viable solution to address the two most challenging issues of unbalanced crystallization and $Sn^{2+}$ oxidation in mixed Sn-Pb perovskites and conduct a thorough inves-tigation to gain a comprehensive understanding of the underlying mechanisms. By tailoring the mismatch in crystallization rates between Sn and Pb perovskites and modulating the grain growth and crystal-lization processes, we were able to significantly improve the crystal quality and uniformity of Sn-Pb perovskite films by introducing a multifunctional N-(carboxypheny)guanidine hydrochloride (CPGCl) molecule, also known as 4-guanidinobenzoic acid hydrochloride. Moreover, the introduction of CPGCl molecules established robust coordination bonds with charged defects, increasing the electron cloud density surrounding these defects. This, in turn, resulted in an intrinsic barrier against $Sn^{2+}$ oxidation. Compared to traditional methodologies, our strategy is more effective in preventing the oxi-dation of $Sn^{2+}$ as it functions before, during, and after the formation of Sn-Pb perovskite films, thereby significantly reducing the density of nonradiative recombination centers. Leveraging the systematic effects facilitated by CPGCl, we successfully developed high-performance NBG PSCs using $FA_{0.7}MA_{0.3}Pb_{0.5}Sn_{0.5}I_3$ (FA=formamidinium, MA=me-thylammonium) light absorbers. The best-performing single-junction NBG cell exhibited a stabilized PCE of 23.11%, and the unencapsulated Sn-Pb cell retained 97.45% of its initial efficiency after 3500 h of shelf storage in an $N_2$ atmosphere. Furthermore, by integrating the CPGCl-modified NBG perovskites into two-terminal (2T) monolithic all-perovskite tandem solar cells, we achieved a stabilized PCE of 28.20% and a certified PCE of 27.35%.

## Results

### Molecular design and density functional theory (DFT) analysis
Firstly, we hypothesized that CPGCl could establish strong hydrogen interactions with tin (II) iodide ($SnI_2$), thereby decelerating the rapid crystallization of Sn perovskites and creating an electronic environ-ment unfavorable for oxidation. Secondly, from the prospect of functional groups of CPGCl, we anticipated that the incorporation of CPGCl molecules could bring sufficient defect passivation of films and improve the photovoltaic performance of devices. In the case of mixed Sn-Pb perovskites, Sn and Pb defects often act as nonradiative recombination centers at the surfaces and grain boundaries, and Lewis acid/base materials can realize electronic passivation for such defects[33]. In this work, it is inferred that the C=O groups of CPGCl molecules are able to reduce under-coordinated $Sn^{2+}$ and $Pb^{2+}$ ions and provide electronic passivation through electron pair donation. Addi-tionally, strong hydrogen bonding interactions were expected to occur between the O−H groups and $I^-$. The interactions can introduce nucleation sites, slow down crystal growth, and align the grain orien-tations in the vertical direction[15,34]. Moreover, the hydrogen bonding at the grain boundaries also has the potential to restrict the migration of iodide ions and boost the long-term stability of perovskite devices[34]. Furthermore, $Cl^-$ ions may assist in defect passivation[35,36].

To validate our hypothesis, we performed a DFT analysis to design the molecular structure and clarify the physical mechanism respon-sible for achieving equilibrium crystallization and suppressing $Sn^{2+}$ oxidation in this systematic strategy. In our study, CPGCl molecules were utilized as dopants at a mole ratio of 1% (relative to A site) in Sn-Pb perovskite precursor solutions to modulate the crystallization of Sn perovskites and Pb perovskites. We examined the influence of CPGCl on the crystallization kinetics of mixed Sn-Pb perovskites. Firstly, the binding energies of various phases were studied using DFT analysis, as shown in Fig. 1a. The DFT results revealed strong hydrogen bonds formed among CPG, lead iodide ($PbI_2$), and N,N-dimethylformamide (DMF), facilitating the formation of CPG-based intermediate phases upon the addition of CPG to the precursors. Additionally, the hydro-gen bonds between $PbI_2$ and formamidinium iodide (FAI) were com-paratively weaker than those between $PbI_2$ and CPG molecules, suggesting that CPG molecules took precedence over FAI in bonding with $PbI_2$. The robust intermediate products of $PbI_2$-CPG-DMF and CPG-$PbI_2$ gradually released $PbI_2$ during the film annealing process (Fig. 1b), thus decelerating the crystallization of Pb perovskites and influencing Sn perovskites as well.

Typically, the crystallization of Sn perovskites occurs at a much faster rate compared to that of their Pb counterparts, and this disparity in crystallization rates presents a significant challenge for Sn-Pb mixed perovskites[37]. As illustrated in Fig. 1c, $SnI_2$ was also gradually released along with the decomposition of intermediate compounds, which is similar to $PbI_2$. Upon the addition of CPG to precursor solutions, the emergence of the $SnI_2$-CPG-DMF phase took precedence over the formation of other phases because of its highest binding energy. This phase would disintegrate once sufficient energy was utilized to over-come the energy barrier. However, the interaction between $SnI_2$ and FAI remained hindered due to the stronger hydrogen bonding between CPG and $SnI_2$. Only after the CPG-$SnI_2$ phase was disrupted, $SnI_2$ could bind to FAI and form perovskite phases. The schematic illustrating the CPG-manipulated crystallization process is presented in Fig. 1d. A noteworthy finding was the significantly stronger hydrogen bonding interactions between $SnI_2$ and CPG compared to those between $PbI_2$ and CPG. Additionally, the binding energy of the $SnI_2$-CPG-DMF compounds was considerably higher than that of the $PbI_2$-CPG-DMF compounds. This fact suggests that the intermediate pro-ducts formed by CPG and $SnI_2$ were more stable than the intermediate phases based on CPG and $PbI_2$. Consequently, it can be inferred that $SnI_2$ was released later than $PbI_2$ from these intermediate phases. Therefore, we can deduce that, in comparison with Pb perovskites, CPG molecules had a greater impact on the delay of crystallization and grain growth in Sn perovskites, resulting in a delayed and balanced crystallization process in Sn-Pb mixed perovskites.

Apart from achieving equilibrium crystallization in mixed Sn-Pb perovskites, CPG molecules also exhibited strong electrostatic attrac-tion and coordination interaction with charged defects. This interac-tion enhanced the electron cloud surrounding the defects, leading to improved energy barriers against $Sn^{2+}$ oxidation. The presence of CPG

molecules on the surface of perovskite crystals altered the distribution of charge density, creating an unfavorable environment for the formation of under-coordinated Sn ions and the oxidation of $Sn^{2+}$. Previous studies have shown that the oxidation of $Sn^{2+}$ often occurs because of the adsorption of oxygen molecules ($O_2$) on the surface of Sn-Pb perovskites[29]. DFT analysis exhibited that in the presence of CPG, $O_2$ tended to combine with CPG through hydrogen bonding, resulting in the co-adsorption of $O_2$-CPG. We studied the charge density distribution induced by $O_2$ adsorption and $O_2$-CPG co-adsorption (Fig. 1e, f). The altered charge density had a significant impact on charge transfer. Under $O_2$ adsorption, each surface unsaturated Sn ion tended to lose approximately 0.33 e, while this charge transfer value would be decreased to 0.25 e under an $O_2$-CPG co-adsorption situation, suggesting a visible influence of CPG in suppressing $Sn^{2+}$ oxidation (Fig. 1g). With the presence of CPG, O atoms in $O_2$ had the

potential to gain more electrons due to the synergetic charge transfer processes through O-Sn interaction as well as O-H hydron bonding with CPG. The changed charge density distribution on the perovskite surface could bring an increased energy barrier for Sn oxidation. Furthermore, we deduced that CPG could effectively reduce the under-coordinated $Sn^{2+}$ and $Pb^{2+}$ ions benefitting from the intense interactions between CPG and $SnI_2$, as well as CPG and $PbI_2$. The reduction in under-coordinated $Sn^{2+}$ and $Pb^{2+}$ ions helped hinder the formation of the nonradiative recombination centers, thus facilitating the transport and extraction of charge carriers[38].

### Characterization of mixed Sn-Pb perovskite films

The above DFT analysis suggests that CPG has the capability to improve the film properties of Sn-Pb perovskites by delaying and balancing crystallization, as well as hindering $Sn^{2+}$ oxidation. To

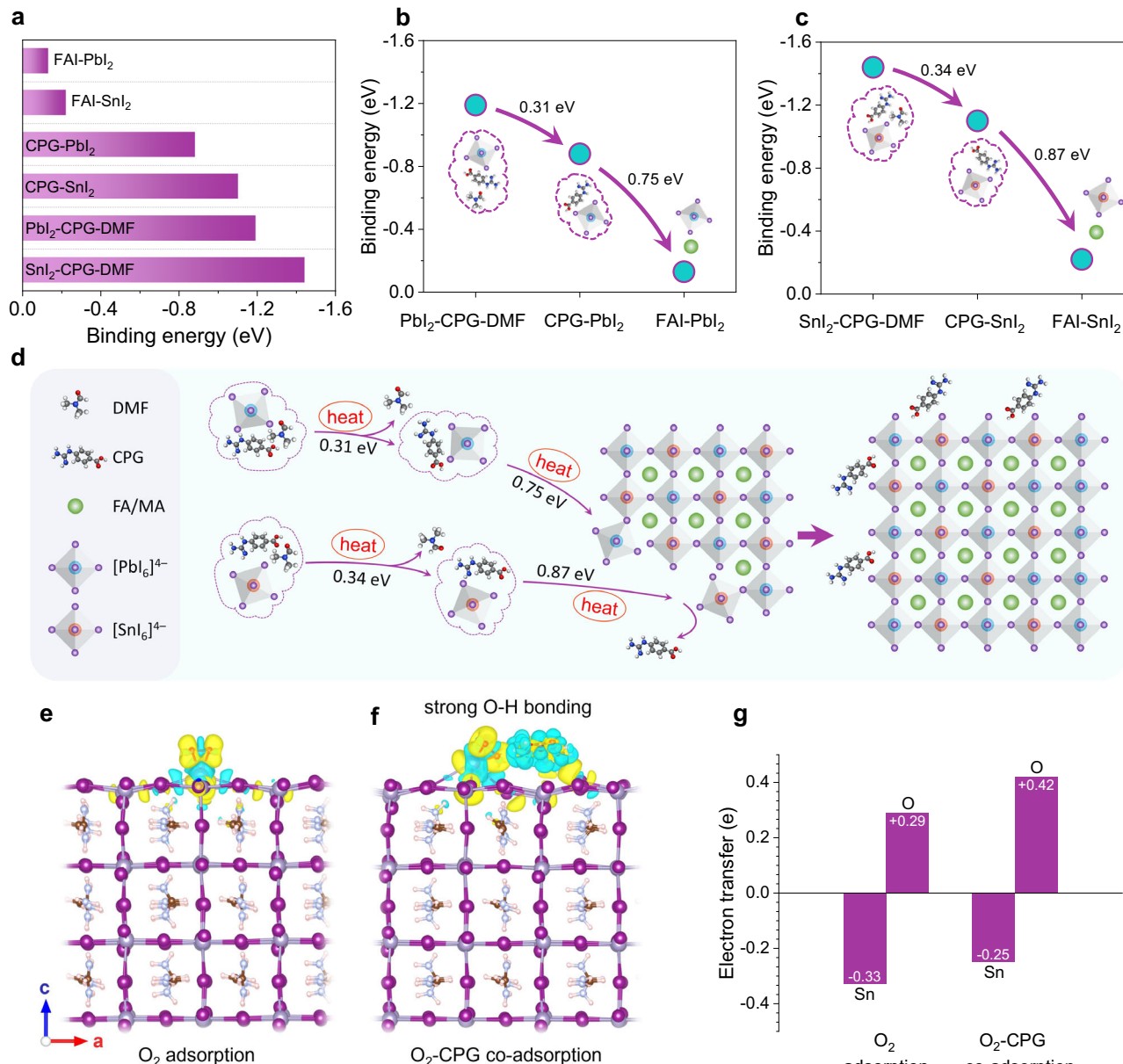

**Fig. 1 | DFT analysis of CPG-assisted crystallization and suppressing $Sn^{2+}$ oxidation. a** Binding energies of FAI-$SnI_2$ ($FASnI_3$), FAI-$PbI_2$ ($FAPbI_3$), and CPG-induced intermediate states. **b** Schematic of the drops in the binding energy of $PbI_2$-CPG-DMF, CPG-$PbI_2$, and FAI-$PbI_2$. **c** Schematic of the drops in the binding energy of $SnI_2$-CPG-DMF, CPG-$SnI_2$, and FAI-$SnI_2$. **d** Schematic of CPG-induced equilibrium

crystallization in mixed Sn-Pb perovskites. **e**, **f** Distributions of charge density induced by $O_2$ adsorption and $O_2$-CPG co-adsorption on the surfaces of Sn-Pb perovskites. **g** Electron transfer of O atoms in $O_2$ and Sn ions in perovskites under $O_2$ adsorption and $O_2$-CPG co-adsorption.

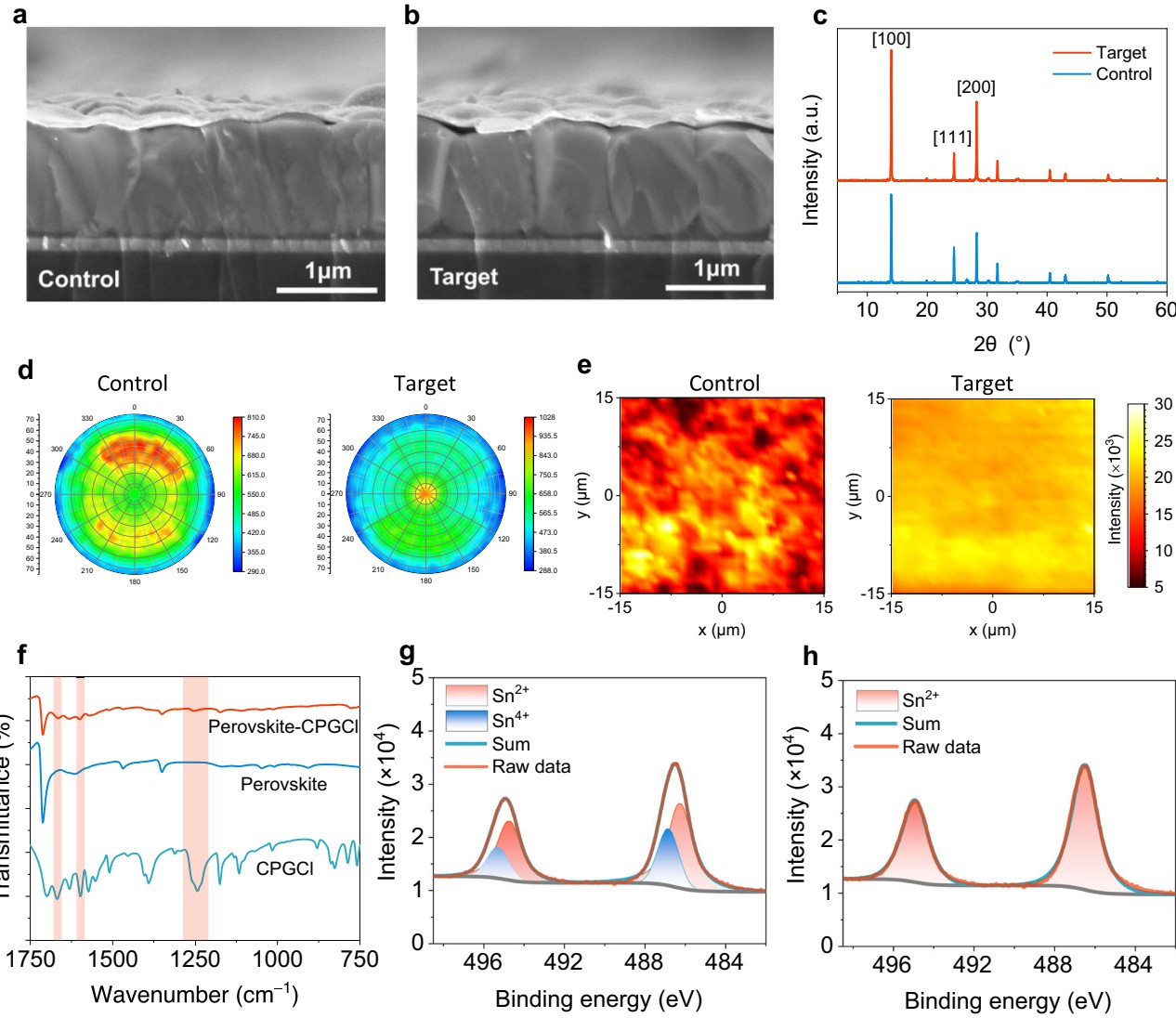

**Fig. 2 | Characterizations of mixed Sn-Pb perovskite films. a, b** Cross-section SEM images of Sn-Pb perovskite films without (**a**) and with (**b**) the presence of CPGCl. **c, d** XRD patterns (**c**) and pole figures (**d**) of control and target Sn-Pb perovskite films. **e** Micro-PL mapping of control and target mixed Sn-Pb perovskite films. **f** FTIR spectra of perovskite, perovskite-CPGCl, and CPGCl. **g, h** XPS spectra of Sn 3*d* of control (**g**) and target (**h**) perovskites.

confirm this, we prepared NBG perovskite films to evaluate the influences of CPGCl on the practical film characteristics. Firstly, we explored the effects of CPGCl molecules on the crystal structure and morphology of Sn-Pb perovskite films. Cross-sectional scanning electron microscopy (SEM) images revealed visible changes in the crystal structure of Sn-Pb perovskite films upon the CPGCl addition, as shown in Fig. 2a, b. In the control perovskite film, longitudinal grain boundaries were present, which could impede the carrier transfer process, particularly in thick light absorbers[39]. By contrast, in the CPGCl-modified film, the grains exhibited more orderliness, growing along the vertical direction. Furthermore, a single grain could span the hole transport layer and the electron transport layer, indicating that manipulating crystallization had positive effects on crystal growth in mixed Sn-Pb perovskites. The improved grain growth and crystal structure facilitated efficient charge-carrier transport and extraction in Sn-Pb perovskite films. Additionally, we found that balanced crystallization had a significant impact on the surface morphology of Sn-Pb perovskite films. The CPGCl-modified perovskite film displayed a smoother surface and a more regular grain distribution without impurities phases at the grain boundaries, as evidenced by the top-view SEM images (Supplementary Fig. 1). The reduction in surface

roughness was further validated via atomic force microscopy (AFM) measurements (Supplementary Fig. 2). The average roughness of Sn-Pb perovskite films decreased from 58.7 to 45.6 nm with the doping of CPGCl. The smoother surfaces facilitated better contact between perovskite films and electron transport layers, enhancing charge-carrier extraction at the interfaces[40]. Moreover, the light absorption efficiency depends on not only the thickness of perovskite films but also the crystallization quality. As shown in Supplementary Fig. 3, the light absorbance of the CPGCl-passivated perovskite film was much higher than that of the control group, even with a similar thickness, indicating much more efficient light harvesting and photon-to-exciton conversion within the CPGCl-passivated film.

To gain further insights into the role of CPGCl agents in the crystal structure, X-ray diffraction (XRD) measurements were performed. The XRD signals corresponding to the [100] and [200] planes exhibited increased intensity, while the signal of the [111] plane weakened, and signals of other planes did not exhibit significant changes after incorporating CPGCl in the films (Fig. 2c). Additionally, in the XRD patterns of CPGCl-modified Sn-Pb perovskite films, no peaks were detected in the low-angle range of 2θ < 10°, indicating the probable absence of significant low-dimensional perovskite phases (Fig. 2c and

Supplementary Fig. 4). The lack of low-dimensional or other impurities phases suggests that the enhancement of film properties was not related to low-dimensional heterostructures in our perovskite films. Furthermore, the XRD measurements revealed that all the compounds induced by CPGCl molecules existed as intermediate phases during the formation of perovskites and eventually decomposed. To further illustrate the strong hydrogen interactions generated by CPGCl molecules, we added an overdose of $PbI_2$ (2 or 4%) to the perovskite precursors and performed XRD tests on the films. With the presence of CPGCl molecules, the XRD signals corresponding to $PbI_2$ were noticeably weakened (Supplementary Fig. 5), indicating an interaction between CPGCl and $PbI_2$. We additionally conducted dipole figure measurements along the [100] facet orientation. As shown in Fig. 2d, the CPGCl-modified Sn-Pb perovskite film exhibited an enhanced [100] facet orientation, promoting vertical grain growth[41,42]. Benefiting from the manipulation of crystallization, the uniformity of Sn-Pb perovskite films was improved, as confirmed by micro-photoluminescence (PL) mapping (Fig. 2e). The micro-PL mapping of the target perovskite film showed brighter and more uniform PL emissions compared to the control film, indicating improved film uniformity and reduced non-radiative recombination[16], which can be attributed to the presence of CPGCl. We further performed time-resolved PL (TRPL) to investigate the charge-carrier dynamics in the films (Supplementary Fig. 6). The target perovskite film with CPGCl molecules showed an average carrier lifetime of 567 ns, which was much longer than that of the control film (55 ns), confirming the passivation effects of CPGCl. The interaction between CPGCl and perovskites was further supported by Fourier transform infrared (FTIR) measurements (Fig. 2f). Several peaks corresponding to CPGCl were identified in the FTIR spectra of the target perovskite film. The C=O stretching vibration peak shifted from 1669 to 1665 $cm^{-1}$, suggesting a change in force constant, which we speculated is related to the coordination between CPGCl and $Pb^{2+}$ or $Sn^{2+}$ at the grain boundaries[15,43]. X-ray photoelectron spectroscopy (XPS) measurements also confirmed the existence of interactions between CPGCl and Sn-Pb perovskites. The observed shifts in Pb 4f and I 3d binding energies following the incorporation of CPGCl molecules suggested that CPGCl molecules are capable of passivating under-coordinated $Pb^{2+}$ and $I^-$ defects[44] (Supplementary Fig. 7).

There is relatively weak coulombic attraction between electrons in s orbitals and the nucleus of $Sn^{2+}$, thus allowing for the facile conversion of $Sn^{2+}$ into $Sn^{4+}$[26]. The oxidation of $Sn^{2+}$ occurs before, during, and after the formation of Sn-Pb perovskites, resulting in a high trap density and inadequate carrier extraction[15]. We observed a color transformation in the pristine perovskite precursor solutions, comprising FAI, methylammonium iodide (MAI), $SnI_2$, $PbI_2$, and $SnF_2$ dissolved in a mixture of DMF and dimethylsulfoxide (DMSO) solvents, shifting from yellow to dark red upon exposure to ambient air (Supplementary Fig. 8). This alteration indicated the formation of $Sn^{4+}$ resulting from the oxidation of $Sn^{2+}$. As expected, the color change was noticeably decelerated in the CPGCl-doped precursor solutions under the same conditions. The suppression of $Sn^{2+}$ oxidation brought by CPGCl doping was further supported by Sn 3d spectra obtained from XPS tests. In the perovskite films without CPGCl, the ratio of $Sn^{4+}/Sn^{2+}$ determined from the integrated peak proportions of $Sn^{4+}$ and $Sn^{2+}$ was approximately 45%, indicating obvious $Sn^{2+}$ oxidation (Fig. 2g). By contrast, the oxidation of $Sn^{2+}$ was almost eliminated when CPGCl molecules were introduced into the perovskite films (Fig. 2h), consistent with the results of DFT analysis. On the basis of the aforementioned characterizations and DFT analysis, we deduced that CPGCl could effectively inhibit the oxidation of $Sn^{2+}$ before, during, and after the preparation of Sn-Pb mixed perovskite films, therefore reducing trap density and nonradiative recombination centers and ultimately promoting efficient charge collection[30].

To further investigate the influence of CPGCl on film properties, the effects of passivating agents on energy levels were examined using ultraviolet photoemission spectroscopy (UPS), and the results are plotted in Supplementary Fig. 9. When the perovskites were amended by CPGCl, both the highest occupied molecular orbital (HOMO) and lowest unoccupied molecular orbital (LUMO) shifted toward shallower energy states, which was beneficial for hole transport. By comparing the distance between the LUMO and the Fermi levels ($E_F$) with the distance between the HOMO and the $E_F$, we observed that the target perovskites had a more n-type semiconducting nature, likely due to the suppression of p-type self-doping arising from the oxidation of $Sn^{2+}$[45,46]. Note that the control perovskite also presented n-type semiconducting features, which may be attributed to the post-treatment modification with ethane-1,2-diammonium iodide (EDAI$_2$)[47]. The band alignment of both control and target perovskite films is shown in Supplementary Fig. 10. Note that the band-gaps of the perovskites without and with CPGCl doping, evaluated using Tauc plots, showed similar results of approximately 1.26 eV (Supplementary Fig. 11).

## Photovoltaic characteristics of single-junction Sn-Pb PSCs

Motivated by the improved film properties observed with the addition of CPGCl, we proceeded to investigate its effects on device performance. We fabricated planar single-junction Sn-Pb PSCs employing an inverted p–i–n configuration of indium tin oxide (ITO)/Poly(3,4-ethylenedioxythiophene) polystyrene sulfonate (PEDOT:PSS)/NBG (1.26 eV) perovskite/$C_{60}$/bathocuproine (BCP)/Cu, as illustrated in Fig. 3a. To ensure adequate light absorption in the near-infrared region, precursor solutions with a high concentration of 2.4 M were utilized to prepare NBG perovskite films. To introduce the passivating agent into the Sn-Pb perovskite films, 1 mol% of CPGCl was directly added to the perovskite precursor solutions. The target group, composing PSCs with CPGCl-doped perovskite films, demonstrated significantly improved photovoltaic performance. As presented in Fig. 3b, the CPGCl modification of the perovskite films led to significant improvements in the photovoltaic parameters: the open-circuit voltage ($V_{OC}$) increased from 0.83 to 0.87 V, the short-circuit current density ($J_{SC}$) increased from 30.82 to 32.22 mA cm$^{-2}$, and the fill factor (FF) increased from 78.13 to 79.11%. We further presented the statistical distribution of PCE for both control and target devices in Fig. 3c, with the averaged PCE elevated from $19.95 \pm 0.37\%$ to $22.53 \pm 0.48\%$ thanks to the incorporation of CPGCl. The statistics of $V_{OC}$, $J_{SC}$, and FF are also provided in Supplementary Fig. 12. Figure 3d depicts the J-V curves of the best-performing single-junction mixed Sn-Pb cell with the following photovoltaic parameters: PCE of 23.15%, $V_{OC}$ of 0.88 V, $J_{SC}$ of 32.77 mA cm$^{-2}$, and FF of 80.11% when measured under a reverse voltage scan (Table 1). The substantial improvement in $J_{SC}$ is particularly crucial as it directly influences the matching of photocurrent densities in all-perovskite tandem solar cells. The stabilized PCE of the best-performing Sn-Pb cell reached 23.11% (Fig. 3e). The high $J_{SC}$ of our CPGCl-modified PSCs was supported by external quantum efficiency (EQE) measurements. As shown in Fig. 3f, the CPGCl-modified cell exhibited an integrated $J_{SC}$ of 32.33 mA cm$^{-2}$, consistent with the values obtained from the J-V measurements. In addition to efficiency, CPGCl also significantly improved device stability. A comparison of the shelf stability of NBG perovskite cells without and with CPGCl addition is shown in Fig. 3g. The PCE of the control device dropped to 79% of its initial value after 800 h of shelf storage in an N$_2$-filled glovebox. By contrast, the target device retained 97.45% of its original PCE even after undergoing an extended long-term test lasting 3500 h, implying a significant improvement in stability through the CPGCl doping in mixed Sn-Pb perovskites. Additionally, we explored the operational stability of unencapsulated devices by subjecting them to aging through maximum power point tracking under 1-sun illumination in an N$_2$-filled glovebox at ~55 °C. As depicted in Supplementary Fig. 13, the PCE of the control device declined to 90% of its initial value after ~53 h, while the target device maintained 90% of its initial PCE after ~128 h.

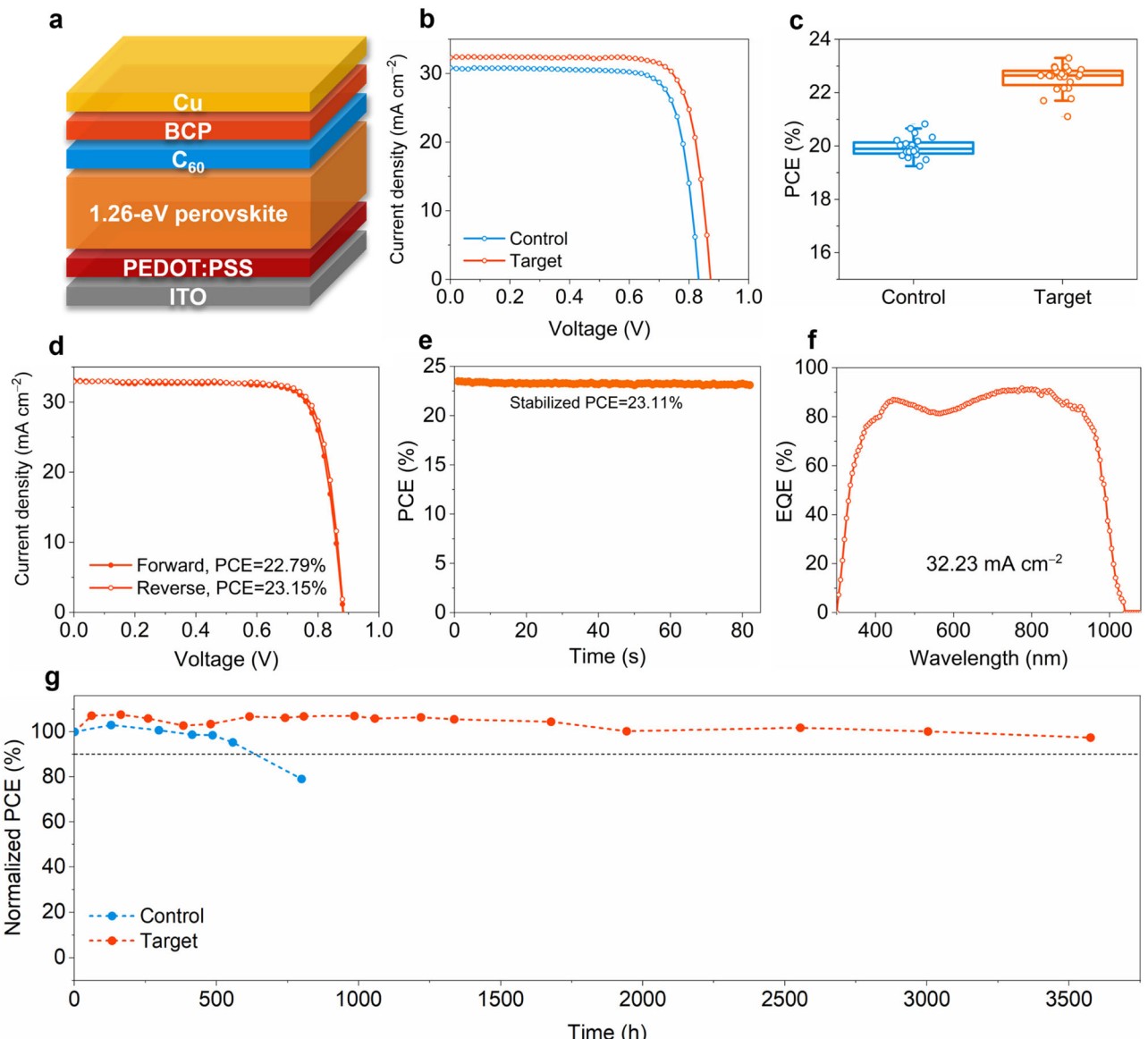

**Fig. 3 | Performance of single-junction Sn-Pb PSCs. a** Schematic of inverted p–i–n single-junction Sn-Pb PSCs. **b** J-V curves of typical control and target single-junction Sn-Pb PSCs. **c** Statistics of PCE for control and target devices, 24 devices for each type. **d**, **e** J-V curves (**d**) and stabilized PCE (**e**) of the best-performing Sn-Pb PSC. **f** EQE curve of the Sn-Pb cell. **g** The long-term stability of unencapsulated control and target single-junction Sn-Pb cells stored in an $N_2$-filled glovebox.

Further enhancements in device stability could involve the use of solid encapsulation and stable charge transport layers.

The improved device performance can be primarily attributed to the reduction in carrier recombination, as confirmed by electrochemical impedance spectra (EIS) and dark J-V measurements. As shown in Supplementary Figs. 14, 15, the Sn-Pb perovskite devices modified with CPGCl exhibited a significant increase in recombination resistance ($R_{rec}$)

### Table 1 | Photovoltaic parameters of the best-performing NBG subcell and all-perovskite tandem solar cell

| Device | Scan direction | $V_{OC}$ (V) | $J_{SC}$ (mA cm$^{-2}$) | FF (%) | PCE (%) |
|---|---|---|---|---|---|
| NBG subcell | Forward | 0.88 | 32.99 | 78.46 | 22.79 |
| | Reverse | 0.88 | 32.77 | 80.11 | 23.15 |
| | Stabilized | - | - | - | 23.11 |
| Tandem cell | Forward | 2.08 | 16.69 | 79.94 | 27.80 |
| | Reverse | 2.11 | 16.65 | 80.23 | 28.24 |
| | Stabilized | - | - | - | 28.20 |

and a marked decrease in leakage current, suggesting enhanced photovoltaic properties achieved through efficient grain passivation and reduced charge-carrier recombination in the devices. Additionally, we investigated the influence of bias voltage on the capacitance of the devices (Supplementary Fig. 16). For the perovskite cells treated with CPGCl, the built-in potential ($V_b$) increased, indicating an enhanced electric field within the devices, which agrees well with the improved $V_{OC}$ observed in the target devices. Nonradiative recombination losses hurt the $V_{OC}$ of PSCs[38], which can be investigated by analyzing the dependence of $V_{OC}$ on light intensity. To access this, we studied the ideality factors of control and target mixed Sn-Pb PSCs by testing the $V_{OC}$ under varying light intensities (Supplementary Fig. 17). The ideality factor of the target device was found to be 1.35, which was smaller than that of the control device (1.39), indicating a reduction in nonradiative recombination induced by CPGCl[48]. We further investigated the influence of CPGCl on trap-state density by performing space-charge limited-current (SCLC) measurements for hole-only devices. As trap-state density is directly proportional to the trap-filled limited voltage ($V_{TFL}$)[29], the decrease of $V_{TFL}$ indicated a reduction in trap density (Supplementary

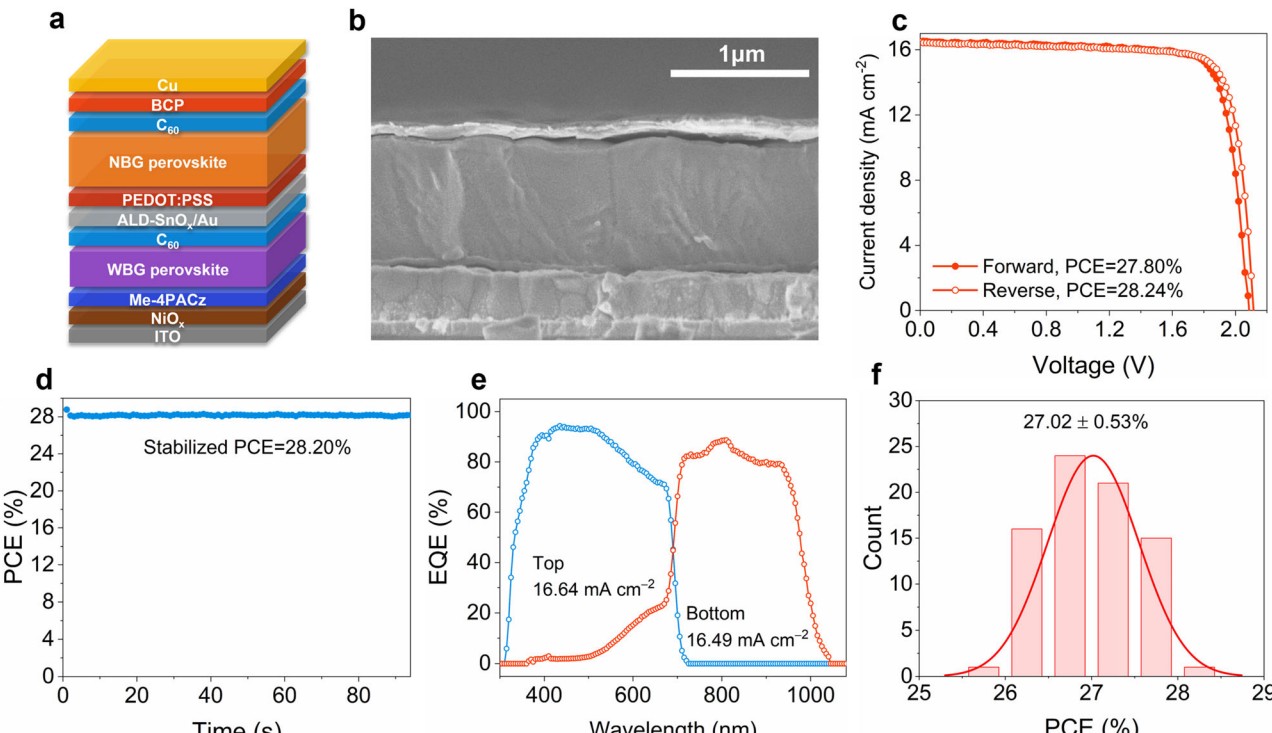

**Fig. 4 | Performance of 2T all-perovskite tandem solar cells with CPGCl doping. a**, **b** Schematic picture (**a**) and cross-section SEM image (**b**) of all-perovskite 2T tandem solar cells. **c**, **d** J-V curves (**c**) and stabilized PCE (**d**) of the best-performing all-perovskite tandem cell. **e** EQE curves of the tandem cell. **f** Histogram of PCEs for 78 tandem cells, the average value was 27.02 ± 0.53%.

Fig. 18). The reduced trap density is beneficial for efficient carrier extraction and transport, aligning with the enhanced photovoltaic performance in CPGCl-modified devices.

### Photovoltaic performance of all-perovskite tandem cells

Taking inspiration from the exceptional performance of single-junction Sn-Pb cells, we then proceeded to fabricate all-perovskite tandem cells utilizing CPGCl-passivated $FA_{0.7}MA_{0.3}Pb_{0.5}Sn_{0.5}I_3$ perovskites for the bottom NBG subcells. The configuration of 2T all-perovskite tandem cells is as follows: ITO/NiO$_x$/[4-(3,6-dimethyl-9H-carbazol-9-yl)butyl]phosphonic acid (Me-4PACz)/WBG (1.77 eV) perovskite/C$_{60}$/ALD SnO$_x$/Au/PEDOT:PSS/NBG perovskite/C$_{60}$/BCP/Cu (Fig. 4a). A representative cross-sectional SEM image of the tandem cells is displayed in Fig. 4b. Figure 4c presents the J-V curves of the best-performing all-perovskite tandem cell, demonstrating a PCE of 28.24%, $V_{OC}$ of 2.11 V, $J_{SC}$ of 16.65 mA cm$^{-2}$, and FF of 80.23% under a reverse voltage scan (Table 1). The corresponding tandem cell also exhibited a stabilized PCE of up to 28.20% (Fig. 4d), which closely matched the PCE obtained from the J-V measurements with a reverse voltage scan. An independent laboratory also certified an efficiency of 27.35% with a $V_{OC}$ of 2.13 V, $J_{SC}$ of 16.27 mA cm$^{-2}$, and FF of 78.94% (Supplementary Fig. 19). The $J_{SC}$ values of the top WBG subcell and the bottom NBG subcell, integrated from the EQE curves, were 16.64 and 16.49 mA cm$^{-2}$, respectively (Fig. 4e), aligning well with the $J_{SC}$ values obtained from the J-V measurements. Furthermore, the tandem cells demonstrated superior long-term stability, as evidenced by maintaining 95.7% of their initial PCE after being stored in an N$_2$-filled glovebox for 2200 h (Supplementary Fig. 20). The J-V curves of our best-performing NBG cell, WBG cell, and all-perovskite tandem cell are plotted in Supplementary Fig. 21. For statistical analysis, we produced a total of 78 individual all-perovskite tandem cells, yielding an average PCE of 27.02 ± 0.53% (Fig. 4f). The statistics of $V_{OC}$, $J_{SC}$, FF, and PCEs are also provided in Supplementary Fig. 22. Despite significant improvements, future studies should focus on enhancing the intrinsic stability

of perovskite subcells, particularly the NBG subcells. The current limitations on the lifetime of tandem cells highlight the necessity for further stabilization of subcells. Additionally, despite achieving high efficiencies in individual subcells, the PCE of all-perovskite tandem cells lags significantly behind their theoretical limits. To fully exploit the potential of high-efficiency subcells, research is required to develop stable interconnection layers that reduce both electrical and optical losses in tandem cells[49].

## Discussion

This work has demonstrated high-performance single-junction Sn-Pb PSCs and all-perovskite tandem solar cells through a systematic strategy that leverages equilibrium crystallization and suppresses Sn$^{2+}$ oxidation in mixed Sn-Pb perovskites. The introduction of CPGCl resulted in intermediate phases that effectively slowed down the crystallization of Sn-Pb perovskites by gradually releasing SnI$_2$ and PbI$_2$ sources. Additionally, the strong coordination between CPGCl molecules and charged defects enhanced the electron cloud density around the defects, leading to significant barriers against Sn$^{2+}$ oxidation. These improvements in film crystallization and reduction of trap density contributed to achieving a stabilized PCE of 23.11% in single-junction NBG PSCs. Additionally, the CPGCl-modified Sn-Pb cells retained 97.45% of their initial PCE after 3500 h of shelf storage in an N$_2$-filled glovebox. Furthermore, the 2T all-perovskite tandem solar cells fabricated using the CPGCl-modified bottom Sn-Pb subcells achieved a certified PCE of 27.35%. This strategy, which focuses on balancing crystallization and restraining Sn$^{2+}$ oxidation in mixed Sn-Pb perovskites, presents a promising pathway for the advancement of perovskite photovoltaic devices.

## Methods

### DFT calculations

DFT calculations were performed using the Vienna ab initio simulation package (VASP)[50,51] with the standard frozen-core projector

augmented-wave (PAW) method[52,53]. The cutoff energy for basis functions was 500 eV. The generalized gradient approximation (GGA) of the Perdew–Burke–Ernzerh (PBE) functional was used for exchange correlation[54]. The Grimme's DFT-D3 scheme was employed for the inclusion of van der Waals interactions[55]. The convergence criteria of the energy difference were set to $1.0 \times 10^{-5}$ eV. All atoms were relaxed until the Hellmann–Feynman forces on them were below 0.01 eV/Å. For bulk calculations, the k-point meshes were chosen such that the product of the number of k-points and corresponding lattice parameters were at least 30 Å. For slab calculations, the Γ-only k-point mesh was chosen, and dipole correction was also considered.

The slabs were separated by a vacuum layer of 25 Å to avoid potential interaction between adjacent layers. The binding energies of bulks (including $FAPbI_3$, $FASnI_3$, CPG-$PbI_2$, CPG-$SnI_2$, $PbI_2$-CPG-DMF, and $SnI_2$-CPG-DMF) were calculated using the following equations: taking Pb-based compounds as an example, $\Delta E_{FAPbI_3} = E_{FAPbI_3} - E_{PbI_2} - E_{FAI}$, where $E_{FAPbI_3}$, $E_{PbI_2}$, and $E_{FAI}$ are calculated total energies of bulk $FAPbI_3$, $PbI_2$, and crystalline FAI in monoclinic symmetry[56]. $\Delta E_{CPGCl-PbI_2} = E_{CPG-PbI_2} - E_{PbI_2} - E_{CPGCl}$, where $E_{CPG-PbI_2}$ and $E_{CPGCl}$ are total energies of bulk CPG-$PbI_2$ and gas-phase CPGCl. $\Delta E_{PbI_2-CPG-DMF} = E_{PbI_2-CPG-DMF} - E_{PbI_2} - E_{CPG} - E_{DMF}$, where $E_{PbI_2-CPG-DMF}$ and $E_{DMF}$ are total energies of bulk $PbI_2$-CPG-DMF and gas-phase DMF. The visualization of crystal structures was done using VESTA software[57]. Part of the post-processing was using VASPKIT[58]. Charge analysis was done with a Bader code[59].

## Materials

All the materials were used as received without purification. FAI, methylammonium iodide (MAI), lead bromide ($PbBr_2$), and $SnI_2$ were purchased from Advanced Election Technology Co., Ltd. $PbI_2$ was purchased from TCI. PEDOT:PSS aqueous solutions, BCP, and $C_{60}$ were purchased from Xi'an Polymer Light Technology. $EDAI_2$, $SnF_2$, cesium iodide (CsI), isopropanol (IPA), DMF, DMSO, and chlorobenzene were purchased from Sigma-Aldrich. CPGCl was purchased from Bidepharm. Patterned ITO substrates (12 Ω per square) with a dimension of 20 × 20 mm purchased from Advanced Election Technology Co., Ltd. were used in device fabrication.

## NBG $FA_{0.7}MA_{0.3}Pb_{0.5}Sn_{0.5}I_3$ perovskite precursor solutions

The precursor solutions (2.4 M) were prepared by dissolving $PbI_2$, $SnI_2$, MAI, and FAI in a mixed solvent of DMF and DMSO at the volume ratio of 3:1. For the Sn-rich environment in precursors, $SnF_2$ (10 mol% with respect to $SnI_2$) was added to precursor solutions. The solutions were then stirred for 1 h at room temperature. Finally, the solutions were filtered with 0.22 μm polytetrafluoroethylene (PTFE) membrane before fabricating perovskite films.

## WBG $FA_{0.8}Cs_{0.2}Pb(I_{0.6}Br_{0.4})_3$ perovskite precursor solutions

The precursor solutions (1.2 M) were prepared by dissolving CsI, FAI, $PbBr_2$, and $PbI_2$ in a mixed solvent of DMF and DMSO at a volume ratio of 4:1. The solutions were stirred for 1 h at a temperature of 60 °C and then filtered with 0.22 μm poly(vinylidene fluoride) membrane before use.

## NBG Sn-Pb perovskite solar cell fabrication

ITO substrates were washed in acetone and ethyl alcohol for 15 min respectively, then the plasma treatment was conducted for further cleaning. PEDOT:PSS aqueous solutions were spin-coated on the ITO substrates at a speed of 5000 rpm for 30 s and subsequently annealed on a hotplate at 140 °C for 20 min in ambient air. After cooling, the ITO substrates were transferred into an $N_2$-filled glovebox for fabricating perovskite films. The perovskite films were deposited through a two-step spin-coating utilizing a spin coater (EZ6-S, JIANGSU LEBO SCIENCE INSTRUMENTS CO., LTD) with the following parameters: (1) 1000 rpm for 10 s with an acceleration of 200 rpm s⁻¹ and (2) 4000 rpm for 40 s at an acceleration of 1000 rpm s⁻¹. About 400 μl of

chlorobenzene was dropped onto the spinning ITO substrates during the second step of spin-coating at the thirtieth second after the start. The ITO substrates were then put on a hotplate and annealed at 100°C for 10 min. The post-treatment was performed by depositing the solutions of $EDAI_2$ in IPA with a concentration of 0.5 mg ml⁻¹ on the perovskite films at a speed of 4000 rpm for 30 s, then the substrates were annealed at 100 °C for 5 min. Finally, 20 nm of $C_{60}$, 7 nm of BCP, and 100 nm of Cu were deposited on the perovskite films in sequence via thermal evaporation (Wuhan PDVacuum Technologies Co., Ltd).

## All-perovskite tandem solar cell fabrication

ITO substrates were cleaned as described above. Then, $NiO_x$ nanocrystal layers were fabricated by spin-coating a solution of 10 mg ml⁻¹ $NiO_x$ in pure water at 3000 rpm for 30 s and annealed at 100 °C for 10 min in air. In the following, the self-assembled monolayers of Me-4PACz (0.3 mg ml⁻¹ in ethanol) were spin-coated on the ITO substrates at 3000 rpm for 30 s, and the substrates were heated at 100 °C for 10 min in an $N_2$-filled glovebox. For the fabrication of perovskite films, 50 μl of WBG perovskite precursor solutions was dropped onto each ITO substrate and spin-coated at 5000 rpm for 60 s utilizing a spin coater (EZ6-S, JIANGSU LEBO SCIENCE INSTRUMENTS CO., LTD). About 300 μl of diethyl ether was dropped at 20 s before the end of the spinning process and the substrates were heated at 50 °C for 2 min and 100 °C for 10 min afterward. After the substrates were cooled, post-treatments with 1,3-propane-diammonium iodide (PDA) were conducted via spin-coating a solution of 2 mg ml⁻¹ $PDAI_2$ in IPA at 4000 rpm for 30 s, followed by annealing at 100 °C for 5 min. After cooling, the substrates were transferred to an evaporation system, and a $C_{60}$ film (18 nm) was deposited onto the WBG perovskites through thermal evaporation (Wuhan PDVacuum Technologies Co., Ltd). ALD $SnO_x$ layers with a thickness of 20 nm were deposited on the WBG perovskite films, after which 0.8 nm of Au was deposited via thermal evaporation. The substrates were then transferred to an $N_2$-filled glovebox for the fabrication of NBG films. The NBG films were deposited and treated as described above. Finally, $C_{60}$ (20 nm), BCP (7 nm), and Cu (100 nm) were deposited on the top of NBG perovskite films in order (Wuhan PDVacuum Technologies Co., Ltd).

## Film and device characterization

*J-V* characteristics of devices were obtained using a Keithley 2400 source meter under AM 1.5 G illumination in an $N_2$-filled glovebox. The *J-V* tests and shelf storage of perovskite solar cells were conducted at a temperature of ~26 °C. The light intensity was set at 100 mW cm⁻² and calibrated with a certified WPVS standard solar reference cell (SRC-2020, Enlitech; traceable to NREL) before measurements. For *J-V* scans of mixed Sn-Pb perovskite solar cells, the scanning rate was 0.02 V s⁻¹, the voltage step was 50 mV, and the delay time was 25 ms. The forward scan was from −0.1 to 1 V and the reverse scan was from 1 to −0.1 V. For *J-V* scans of all-perovskite tandem solar cells, the scanning rate was 0.02 V s⁻¹, the voltage step was 50 mV, and the delay time was 25 ms. The forward scan was from −0.1 to 2.2 V and the reverse scan was from 2.2 to −0.1 V. The area of the tested solar cells was 0.0948 cm², and an aperture shade mask was put in front of the solar cells to ensure the active area was 0.070225 cm². EQE data was obtained using a QE/IPCE system (Enli Technology Co., Ltd) by focusing monochromatic light, which was calibrated by a standard silicon photodiode in advance, on device pixels in ambient air.

The crystal characterizations of perovskites, including XRD and pole figure measurements, were performed with an XRD instrument (Bruker AXS, D8 Advance). XPS tests were conducted with a photoelectron spectrometer (Thermo Scientific, ESCLAB 250Xi, USA). EIS spectra were given by a CHI 770E electrochemical workstation

(Shanghai Chenhua Instruments, China) within a frequency range of 1 MHz to 1 Hz. Top-view and cross-section SEM images were gained with a TESCAN AMBER microscope. FTIR spectra were acquired using a NICOLET iS50 FTIR spectrometer. Absorption spectra were recorded by a SHIMADZU mini 1280 UV-visible spectrophotometer. AFM images were given by an AFM instrument (SPM-9500J3, Shimadzu, Japan).

The PL mapping was conducted with a home-built system. The excitation source (PDL 800-D) was a 405 nm pulsed diode laser purchased from Picoquant. The excitation light was focused onto the sample surface using a ×100 objective lens. The on-sample diameter was about 400 nm. The samples were placed on an X-Y motorized stage purchased from Thorlabs. The scanning step size was 1 μm in both the X and Y directions. The emitted light was collected by a spectrometer (Kymera 328b-B1) purchased from Andor.

## Reporting summary

Further information on research design is available in the Nature Portfolio Reporting Summary linked to this article.

## Data availability

All the main data are available in the main text, the Supplementary Information, and the Source Data file. All other data of this study are available from the corresponding authors on request. Source data are provided with this paper.

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

## Acknowledgements

This work is technically supported by the Key Lab of Artificial Micro-and Nano-Structures of the Ministry of Education of China, Wuhan University. The authors thank the Core Facility of Wuhan University for SEM, FTIR, XRD, XPS, and UPS measurements. The authors thank Dr. Ying Zhang and Dr. Mingyuan Du from the Core Facility of Wuhan University for their help with SEM/FIB and FTIR analysis, respectively. This work is financially supported by the National Natural Science Foundation of China (12174290, 12134010, W.K. and 12104345, W.M.) and the Knowledge Innovation Program of Wuhan-Shuguang Project (Grant Number: 2023010201020245, W.K.).

## Author contributions

W.K. and J.Z. conceived the idea and directed the overall project. J.Z. fabricated NBG PSCs and all-perovskite tandem cells and performed film and device characterizations. S.F. helped with the fabrication of WBG subcells in all-perovskite tandem cells. W.M. carried out the DFT analysis. L.H. assisted in UPS, XPS, and EIS measurements. S.Z., C.W. (Cheng Wang), D.P., C.W. (Chen Wang), H.C., H.G., and T.W. offered help in device fabrication and film characterizations. J.Z., W.M., and W.K. wrote the manuscript. All authors discussed the results and contributed to the revisions of the manuscript. G.F. and W.K. supervised this study.

## Competing interests

The authors declare no competing interests.
