## [Peer Review File · Nature Communications]

Mixed tin-lead perovskites with balanced crystallization and oxidation barrier for all-perovskite tandem solar cellsREVIEWER COMMENTS

Reviewer #1 (Remarks to the Author):

In this study, the authors introduced N-(carboxyphenyl)guanidine hydrochloride (CPGCI) into the perovskite precursor solution to serve two purposes; retard Pb-Sn crystallization and reduce trap density to suppress Sn²⁺ oxidation. Pb-Sn-based single-junction and 2T all-perovskite tandem devices with efficiencies of 23.15% and 28.24% were demonstrated. Additionally, the long-term shelf-stability of the target devices showed significant enhancements compared to the control devices in both the single-junction and tandem configurations. The findings of this work are interesting, and I recommend its publication in Nature Communications once the points are acted upon:

1. If the electron cloud density surrounding defects increases, the typical expectation in XPS is that the binding energy would shift to lower values. However, in Supplementary Figure 6, Pb 4f and I 3d core energy levels shifted to higher binding energy levels.
2. In line 204, “allowing for the facile conversion of Sn²⁺ into Sn⁴⁺¹⁸” the subscript 18 needs to be removed.
3. The authors claim that the color of the perovskite precursor solution after being exposed to ambient air maintained its original color only after the addition of CPGCI. However, it is unclear how long the solution was stable for.
4. The authors need evidence of the claim that the incorporation of CPGCI molecules leads to sufficient defect passivation of perovskite films. Techniques such as time-resolved photoluminescence (TRPL), photoluminescence quantum yield (PLQY), or space charge-limited current (SCLC) would be useful.
5. In line 212, it was mentioned that “In the perovskite films without CPGCI, the ratio of Sn⁴⁺/Sn²⁺ determined from the integrated peak proportions of Sn⁴⁺ and Sn²⁺ was approximately 45%, indicating obvious Sn²⁺ oxidation. It is unclear whether this film was fresh or aged. With the assumption that it is a fresh film, the amount of oxidation is exceptionally high. Why? Also, would integrating CPGCI into the perovskite film make it less prone to oxidation when aging the film?”
6. The authors included the long-term shelf-storage stability comparison between control and target devices. The devices must be subjected to harsher conditions to shed light on how effective this strategy is and compare that with what has already been reported in the literature. The operating stability of the devices at MPP, under elevated temperature, is required.

Reviewer #2 (Remarks to the Author):

In this study, Zhou et al. employed N-(carboxyphenyl)guanidine hydrochloride (CPGCI) to facilitate homogenized crystallization and mitigate oxidation in Sn-Pb perovskite solar cells. Employing various characterization techniques, the authors demonstrated the efficacy of this strategy. The optimized devices achieved efficiencies, reaching 23.15% for single-junction Sn-Pb perovskite solar cells and an impressive 28.2% for all-perovskite solar cells. The challenges related to rapid crystallization and susceptibility to oxidation are significant concerns in the community, and this work offers a promising solution with broad potential impact. Overall, this work is of great interest and significance to the community. Therefore, I would recommend publication of this work in Nature Communications after addressing the following issues and comments.

- (1) Accurate matching of solar simulator spectra for narrow-bandgap solar cells and tandems is crucial. Could the authors provide details on the mismatch factor for the devices? The authors should also provide more details on the measurements for tandem solar cells.
- (2) Perovskite solar cells often exhibit J-V curve hysteresis, leading to varied efficiencies from different J-V scanning directions. To ensure accuracy, I recommend reporting only stabilized PCE in the abstract.
- (3) In Figure 2c, the intensity of the (100) and (200) planes for the target films was enhanced. Could the authors also analyze and discuss changes in other planes?
- (4) The authors provided a certified efficiency for their solar cells. It is essential to also provide other photovoltaic parameters in the main text, particularly current density, for comparison within the lab.
- (5) Additional details, such as temperature and bias, in stability tests (Figure 3g, Supplementary Figure 17) for tandem solar cells should be provided.
- (6) Given the growing significance of two-terminal all-perovskite tandem cells, a discussion on current limiting factors and future development routes in this field would enrich the manuscript.
- (7) Supplementary Figure 3 reveals enhanced absorption in the target film compared to the control despite similar thickness (as shown in Figure 2a-b). Could the authors elucidate the underlying reasons for this discrepancy?

Responses to Reviewers

We sincerely appreciate the reviewers for dedicating their time and effort to evaluate our manuscript. Taking into consideration the valuable comments provided, we have diligently revised the manuscript. All suggestions have been incorporated, and each comment has been addressed in our revised version. Additionally, we have included extra experimental results and supplementary explanations to enhance the persuasiveness of our conclusions.

A detailed point-by-point response to the reviewers is provided below. All modifications in the revised manuscript and supporting information are clearly indicated using red font.

Responses to the comments of Reviewer #1

General comment: In this study, the authors introduced N-(carboxyphenyl)guanidine hydrochloride (CPGCl) into the perovskite precursor solution to serve two purposes; retard Pb-Sn crystallization and reduce trap density to suppress Sn^{2+} oxidation. Pb-Sn-based single-junction and 2T all-perovskite tandem devices with efficiencies of 23.15% and 28.24% were demonstrated. Additionally, the long-term shelf-stability of the target devices showed significant enhancements compared to the control devices in both the single-junction and tandem configurations. The findings of this work are interesting, and I recommend its publication in Nature Communications once the points are acted upon:

Response: We would like to express our sincere gratitude to the reviewer for taking the time to conduct a thorough and careful review of our manuscript. We deeply appreciate the feedback provided by the reviewer, and we are grateful for the valuable suggestions that have helped us improve the manuscript.

Comment 1. If the electron cloud density surrounding defects increases, the typical expectation in XPS is that the binding energy would shift to lower values. However, in Supplementary Figure 6, Pb 4*f* and I 3*d* core energy levels shifted to higher binding energy levels.

Response: We thank the reviewer for this comment. Upon reexamining the raw XPS data and retesting the samples, we have confirmed the observed changes in binding energy. In our results, the shifts in XPS curves primarily indicate alterations in the distribution of electric fields and electronic interactions between different ions. Specifically, the change in binding energy is associated with the electronegativity of dopants, resulting in varied binding energy shifts with the introduction of different

molecules. This aligns with existing literature and is consistent with our results, where binding energy shifted to higher values upon the incorporation of some special molecules [Adv. Energy Mater. 11, 2102281 (2021); Adv. Mater. 35, 2300352 (2023)]. As stated in our manuscript, the XPS tests were solely employed to demonstrate the interaction between CPGCl and Pb/I ions. We have revised the corresponding sentences as follows:

“X-ray photoelectron spectroscopy (XPS) measurements also confirmed the existence of interactions between CPGCl and Sn-Pb perovskites. The observed shifts in Pb 4f and I 3d binding energies following the incorporation of CPGCl molecules suggested that CPGCl molecules are capable of passivating under-coordinated Pb²⁺ and I defects⁴⁴ (Supplementary Fig. 7).” (Page 10, Manuscript Draft)

Comment 2. In line 204, “allowing for the facile conversion of Sn²⁺ into Sn⁴⁺18” the subscript 18 needs to be removed.

Response: We value the comprehensive review carried out by the reviewer. Concerning the reference to “18” in line 204, it designates the 18th reference in the manuscript. To prevent any potential confusion, the reference is now cited after the full stop, and the sentence is revised as follows:

“There is relatively weak coulombic attraction between electrons in s orbitals and the nucleus of Sn²⁺, thus allowing for the facile conversion of Sn²⁺ into Sn⁴⁺.²⁶” (Page 10, manuscript draft)

Comment 3. The authors claim that the color of the perovskite precursor solution after being exposed to ambient air maintained its original color only after the addition of CPGCl. However, it is unclear how long the solution was stable for.

Response: We are grateful to the reviewer for highlighting this aspect. In response, we conducted additional tests to offer a more detailed and precise examination of the impact of CPGCl on inhibiting the oxidation of Sn²⁺ in precursor solutions. The revised manuscript and supplementary information files now include updated photographs of precursor solutions exposed to air for varying durations, along with corresponding descriptions.

Supplementary Figure 8. Photographs of control and target Sn-Pb precursor solutions exposed to ambient air for varying durations.

“We observed a color transformation in the pristine perovskite precursor solutions, comprising FAI, MAI, SnI₂, PbI₂, and SnF₂ dissolved in a mixture of DMF and dimethylsulfoxide (DMSO) solvents, shifting from yellow to dark red upon exposure to ambient air (Supplementary Fig. 8). This alteration indicated the formation of Sn⁴⁺ resulting from the oxidation of Sn²⁺. As expected, the color change was noticeably decelerated in the CPGCl-doped precursor solutions under the same conditions.” (page 10, manuscript draft)

Comment 4. The authors need evidence of the claim that the incorporation of CPGCl molecules leads to sufficient defect passivation of perovskite films. Techniques such as time-resolved photoluminescence (TRPL), photoluminescence quantum yield (PLQY), or space charge-limited current (SCLC) would be useful.

Response: We sincerely appreciate the reviewer for this valuable suggestion. In response, we selected both time-resolved photoluminescence (TRPL) and space charge-limited current (SCLC) measurements to validate the efficient defect passivation in CPGCl-modified perovskite films.

The SCLC results have been incorporated into Supplementary Figure 18, accompanied by relevant discussions in the revised manuscript. On the basis of the SCLC results, we confirmed that CPGCl molecules effectively contribute to defect passivation in perovskite films. The TRPL results have been depicted in Supplementary Figure 6. It is apparent that the carrier lifetime of the target film (567 ns) is significantly longer than that of the control film (55 ns), confirming the passivation effects of CPGCl.

Supplementary Figure 18. SCLC curves of control and target hole-only devices.

“We investigated the influence of CPGCl on trap density by performing space-charge limited-current (SCLC) measurements for hole-only devices. As trap density is directly proportional to the trap-filled limited voltage (V_{TFL})²⁹, the decrease of V_{TFL} indicated a reduction in trap density (**Supplementary Fig. 18**). The reduced trap density is beneficial for efficient carrier extraction and transport, aligning with the enhanced photovoltaic performance in CPGCl-modified devices.” (page 14, manuscript draft)

Supplementary Figure 6. Supplementary Figure 6. TRPL spectra of a control film and a target CPGCl-

modified perovskite film. (The average carrier lifetime τ was calculated by $\tau = \frac{A_1\tau_1^2 + A_2\tau_2^2}{A_1\tau_1 + A_2\tau_2}$, where τ_1 and τ_2 denote the decay time constants, and A_1 and A_2 are the corresponding amplitudes of the decay, respectively.)

“We further performed time-resolved PL (TRPL) to investigate the charge-carrier dynamics in the films (**Supplementary Fig. 6**). The target perovskite film with CPGCl molecules showed an average

carrier lifetime of 567 ns, which was much longer than that of the control film (55 ns), confirming the passivation effects of CPGCl.” (pages 9-10, manuscript draft)

Comment 5. In line 212, it was mentioned that “In the perovskite films without CPGCl, the ratio of $\text{Sn}^{4+}/\text{Sn}^{2+}$ determined from the integrated peak proportions of Sn^{4+} and Sn^{2+} was approximately 45%, indicating obvious Sn^{2+} oxidation. It is unclear whether this film was fresh or aged. With the assumption that it is a fresh film, the amount of oxidation is exceptionally high. Why? Also, would integrating CPGCl into the perovskite film make it less prone to oxidation when aging the film?

Response: These are insightful questions. In our XPS measurements, we utilized fresh films. The oxidation of Sn^{2+} takes place before, during, and after the fabrication of mixed Sn-Pb perovskites. Additionally, when the films were transferred into the XPS system, they inevitably came into contact with oxygen molecules, leading to the adsorption of oxygen onto the perovskite film surfaces and exacerbating Sn^{2+} oxidation. XPS analyzes the top atom layers of samples, and while the ratios of $\text{Sn}^{4+}/\text{Sn}^{2+}$ may not be perfectly accurate, the trend for both control and target samples is clearly discernible. The $\text{Sn}^{4+}/\text{Sn}^{2+}$ ratio in our control film aligns with literature values (*Nature Energy*, 2020, 5, 870-880).

Certainly, the integration of CPGCl has demonstrated the ability to suppress film oxidation, as evidenced by our DFT calculation results and supported by our device stability tests (Figure 3g and the newly added Supplementary Figure 13).

Comment 6. The authors included the long-term shelf-storage stability comparison between control and target devices. The devices must be subjected to harsher conditions to shed light on how effective this strategy is and compare that with what has already been reported in the literature. The operating stability of the devices at MPP, under elevated temperature, is required.

Response: We greatly appreciate the constructive advice provided by the reviewer. Following the reviewer's suggestion, we conducted MPP tests with unencapsulated devices in an N_2 -filled glovebox at a temperature of approximately 55 °C. The results have been incorporated into Supplementary Figure 13, accompanied by additional discussions outlined as follows:

“Additionally, we explored the operational stability of unencapsulated devices by subjecting them to aging through Maximum Power Point tracking under 1-sun illumination in an N₂-filled glovebox at approximately 55 °C. As depicted in Supplementary Fig. 13, the PCE of the control device declined to 90% of its initial value after approximately 53 h, while the target device maintained 90% of its initial PCE after approximately 128 hours. Further enhancements in device stability could involve the use of solid encapsulation and stable charge transport layers.” (Pages 13, manuscript draft)

These results substantiate the enhanced stability of the devices following the incorporation of CPGCl. The stability of our control devices, as reported in this study, aligns with our previous work (*Nature*, 2023, 624, 69-73). We acknowledge that direct comparisons of MPP results across different papers can be challenging, given the substantial influence of various factors, including device configuration, encapsulation practices, encapsulation materials, and hole/electron transport materials, among others. While the stability of our CPGCl-incorporated devices may not be outstanding, it still demonstrates substantial improvement compared to the pristine devices. We believe that further advancements in device stability could be achieved through the utilization of solid encapsulation and stable charge transport layers.

Responses to the comments of Reviewer #2

General comment: In this study, Zhou et al. employed N-(carboxyphenyl)guanidine hydrochloride (CPGCl) to facilitate homogenized crystallization and mitigate oxidation in Sn-Pb perovskite solar cells. Employing various characterization techniques, the authors demonstrated the efficacy of this strategy. The optimized devices achieved efficiencies, reaching 23.15% for single-junction Sn-Pb perovskite solar cells and an impressive 28.2% for all-perovskite solar cells. The challenges related to rapid crystallization and susceptibility to oxidation are significant concerns in the community, and this work offers a promising solution with broad potential impact. Overall, this work is of great interest and significance to the community. Therefore, I would recommend publication of this work in Nature Communications after addressing the following issues and comments.

Response: We sincerely thank the reviewer for the thorough and careful reading of our manuscript and appreciate the reviewer for the positive comments and much-valued suggestions.

Comment 1. Accurate matching of solar simulator spectra for narrow-bandgap solar cells and tandems is crucial. Could the authors provide details on the mismatch factor for the devices? The authors should also provide more details on the measurements for tandem solar cells.

Response: We appreciate the reviewer for bringing this to our attention. In response, we have included detailed information on the mismatch factor and measurements for the devices in our revised manuscript.

“The spectral mismatch factors for the certificated wide-bandgap and narrow-bandgap subcells were 1.0068 and 0.9942, respectively.” (please refer to this information in Supplementary Figure 19)

“For J - V scans of all-perovskite tandem solar cells, the scanning rate was 0.02 V s^{-1} , the voltage step was 50 mV , and the delay time was 25 ms . The forward scan was from -0.1 V to 2.2 V and the reverse scan was from 2.2 V to -0.1 V .” (please find this information on Page 20 of our revised manuscript)

Comment 2. Perovskite solar cells often exhibit J - V curve hysteresis, leading to varied efficiencies from different J - V scanning directions. To ensure accuracy, I recommend reporting only stabilized PCE in the abstract.

Response: We thank the reviewer for this valuable suggestion. Corresponding revisions have been accomplished in our revised manuscript as follows:

“Consequently, single-junction Sn-Pb cells exhibited a stabilized power conversion efficiency of 23.11% and maintained 94% of their initial value even after 3500 h of shelf storage in an inert atmosphere without encapsulation. We further integrated CPGCl-amended Sn-Pb perovskites into two-terminal monolithic all-perovskite tandem cells, delivering a stabilized efficiency of 28.20% and a certified value of 27.35%.” (page 2, manuscript draft)

Comment 3. In Figure 2c, the intensity of the (100) and (200) planes for the target films was enhanced. Could the authors also analyze and discuss changes in other planes?

Response: The discussions on the changes of other planes in XRD curves have been incorporated into our revised manuscript as follows:

“The XRD signals corresponding to the [100] and [200] planes exhibited increased intensity, while the signal of the [111] plane weakened, and signals of other planes did not exhibit significant changes after incorporating CPGCl in the films (Fig. 2c).” (pages 8-9, manuscript draft)

Comment 4. The authors provided a certified efficiency for their solar cells. It is essential to also provide other photovoltaic parameters in the main text, particularly current density, for comparison within the lab.

Response: We appreciate this constructive suggestion. In response, more detailed certified photovoltaic parameters have been added to the revised manuscript as follows:

“An independent laboratory also certified an efficiency of 27.35%, V_{oc} of 2.13 V, J_{sc} of 16.27 mA cm⁻², and FF of 78.94% (Supplementary Fig. 19).” (page 14, manuscript draft)

Comment 5. Additional details, such as temperature and bias, in stability tests (Figure 3g, Supplementary Figure 17) for tandem solar cells should be provided.

Response: The additional details, including temperature and bias in stability tests for solar cells, have been included in our revised manuscript as follows:

“The J-V tests and shelf storage of perovskite solar cells were conducted at a temperature of ~26 °C.”
“For J-V scans of mixed Sn-Pb perovskite solar cells, the scanning rate was 0.02 V s⁻¹, the voltage step was 50 mV, and the delay time was 25 ms. The forward scan was from -0.1 V to 1 V and the reverse scan was from 1 V to -0.1 V. For J-V scans of all-perovskite tandem solar cells, the scanning rate was 0.02 V s⁻¹, the voltage step was 50 mV, and the delay time was 25 ms. The forward scan was from -0.1 V to 2.2 V and the reverse scan was from 2.2 V to -0.1 V.” (page 20, manuscript draft)

Comment 6. Given the growing significance of two-terminal all-perovskite tandem cells, a discussion on current limiting factors and future development routes in this field would enrich the manuscript.

Response: We thank the reviewer for this valuable suggestion. Following the reviewer's advice, we have incorporated the related perspective sentences into our revised manuscript as follows:

“Despite significant improvements, future studies should focus on enhancing the intrinsic instability of perovskite subcells, particularly the NBG subcells. The current limitations on the lifetime of tandem cells highlight the necessity for further stabilization of subcells. Additionally, despite achieving high efficiencies in individual subcells, the PCE of all-perovskite tandem cells lags significantly behind their theoretical limits. To fully exploit the potential of high-efficiency subcells, research is required to develop stable interconnection layers that reduce both electrical and optical losses in tandem cells.⁴⁹”
(Pages 15, manuscript draft)

Comment 7. Supplementary Figure 3 reveals enhanced absorption in the target film compared to the control despite similar thickness (as shown in Figure 2a-b). Could the authors elucidate the underlying reasons for this discrepancy?

Response: We appreciate the reviewer for this valuable comment. Light absorption is not only related to the film thickness but also to the film quality. In our case, the enhanced light absorption of the CPGCl-doped perovskite film with a similar thickness can be mainly attributed to the improved

crystalline quality, as demonstrated in Figure 2c.

We thank all the reviewers again for the invaluable and constructive comments, which are very crucial for improving the quality of this work.

REVIEWERS' COMMENTS

Reviewer #1 (Remarks to the Author):

My requests for improvement are now substantially implemented, and the work is ready and appropriate for Nature Comms.

Optionally (and I do not need to rereview for this): The authors might add details about the hole-only devices they fabricated for SCLC, as no information was provided on the structure used, such as which HTLs were used. This suggestion also applies to TRPL; it's unclear whether the perovskite film was deposited on glass or TCO, which would affect data interpretation.

Reviewer #2 (Remarks to the Author):

The authors have addressed the comments and issues raised by the reviewers. I am satisfied with their response and revision. I would recommend acceptance of this manuscript at its current version.

Responses to Reviewers

We sincerely appreciate the reviewers for dedicating their time and effort to evaluate our manuscript. The suggestions have been incorporated, and the comments have been addressed in our revised version. Responses to the reviewers are provided below.

Responses to the comments of Reviewer #1

General comment: My requests for improvement are now substantially implemented, and the work is ready and appropriate for Nature Comms.

Optionally (and I do not need to rereview for this): The authors might add details about the hole-only devices they fabricated for SCLC, as no information was provided on the structure used, such as which HTLs were used. This suggestion also applies to TRPL; it's unclear whether the perovskite film was deposited on glass or TCO, which would affect data interpretation.:

Response: We deeply appreciate the feedback provided by the reviewer and the points highlighted. In response, we have included the configurations of the samples used in the SCLC and TRPL tests in our revised Supplementary Information as follows:

“The samples were fabricated by spin-coating the perovskite precursors directly onto the glass substrates.” (kindly refer to page S8)

“The configuration of samples was ITO/PEDOT: PSS/perovskite/gallium(III) acetylacetonate@poly(3-hexylthiophene)(mass ratio: 1:10)/Au.” (kindly refer to page S20)

Responses to the comments of Reviewer #2

General comment: The authors have addressed the comments and issues raised by the reviewers. I am satisfied with their response and revision. I would recommend acceptance of this manuscript at its current version.

Response: We thank the reviewer for the positive feedback on our revised manuscript. We are pleased that our responses and revisions have addressed the comments and issues raised by the reviewers. The recommendation for acceptance of the manuscript in its current version is greatly appreciated.